# Evidence of Bacterial Community Coalescence between Freshwater and Discharged *tpm*-Harboring Bacterial Taxa from Hospital and Domestic Wastewater Treatment Plants among Epilithic Biofilms

**DOI:** 10.3390/microorganisms11040922

**Published:** 2023-04-02

**Authors:** Rayan Bouchali, Laurence Marjolet, Leslie Mondamert, Teofana Chonova, Sébastien Ribun, Elodie Laurent, Agnès Bouchez, Jérôme Labanowski, Benoit Cournoyer

**Affiliations:** 1UMR Ecologie Microbienne, CNRS 5557, INRAE 1418, Research Group «Bacterial Opportunistic Pathogens and Environment», VetAgro Sup, Aisle 3, 1st Floor, 69280 Marcy L’Etoile, France; 2Institut de Chimie des Milieux et des Matériaux de Poitiers (IC2MP), École Nationale Supérieure d’Ingénieurs (ENSIP), UMR CNRS 7285, Université de Poitiers, 86000 Poitiers, France; 3UMR CARRTEL, INRAE, Université de Savoie Mont Blanc, 75 Avenue de Corzent, 74200 Thonon-les-Bains, France

**Keywords:** environmental risk assessment, pharmaceuticals, epilithic biofilms, DNA metabarcoding, river microbiomes

## Abstract

The ability of WWTP outflow bacteria at colonizing rock surfaces and contributing to the formation of river epilithic biofilms was investigated. Bacterial community structures of biofilms (b-) developing on rocks exposed to treated wastewaters (TWW) of a hospital (HTWW) and a domestic (DTWW) clarifier, and to surface waters of the stream located at 10 m, 500 m, and 8 km from the WWTP outlet, were compared. Biofilm bacterial contents were analyzed by cultural approaches and a *tpm*-based DNA metabarcoding analytical scheme. Co-occurrence distribution pattern analyses between bacterial datasets and eighteen monitored pharmaceuticals were performed. Higher concentrations of iohexol, ranitidine, levofloxacin, and roxithromycin were observed in the b-HTWW while atenolol, diclofenac, propranolol, and trimethoprim were higher in the b-DTWW. MPN growth assays showed recurrent occurrences of *Pseudomonas aeruginosa* and *Aeromonas caviae* among these biofilms. An enrichment of multi-resistant *P. aeruginosa* cells was observed in the hospital sewer line. *P. aeruginosa* MPN values were negatively correlated to roxithromycin concentrations. The *tpm* DNA metabarcoding analyses confirmed these trends and allowed an additional tracking of more than 90 species from 24 genera. Among the recorded 3082 *tpm* ASV (amplicon sequence variants), 41% were allocated to the *Pseudomonas*. Significant differences through ANOSIM and DESeq2 statistical tests were observed between ASV recovered from b-HTWW, b-DTWW, and epilithic river biofilms. More than 500 ASV were found restricted to a single sewer line such as those allocated to *Aeromonas popoffii* and *Stenotrophomonas humi* being strictly found in the b-HTWW file. Several significant correlations between *tpm* ASV counts per species and pharmaceutical concentrations in biofilms were recorded such as those of *Lamprocystis purpurea* being positively correlated with trimethoprim concentrations. A *tpm* source tracking analysis showed the b-DTWW and b-HTWW *tpm* ASV to have contributed, respectively, at up to 35% and 2.5% of the epilithic river biofilm *tpm*-taxa recovered downstream from the WWTP outlet. Higher contributions of TWW taxa among epilithic biofilms were recorded closer to the WWTP outlet. These analyses demonstrated a coalescence of WWTP sewer communities with river freshwater taxa among epilithic biofilms developing downstream of a WWTP outlet.

## 1. Introduction

Wastewaters (WW) can have complex chemical and microbiological contents [1]. The 16S rRNA gene DNA metabarcoding approaches showed domestic WW to harbor significant numbers of *Pseudomonas*, *Acinetobacter*, *Arcobacter*, *Trichococcus*, *Tetrasphaera*, *Rhodoferax*, *Rhodobacter,* and *Hyphomicrobium* [2,3,4]. Similar analyses performed on hospital WW revealed bacterial taxa restricted or in higher numbers in these waters than the domestic ones such as the *Comamonas*, *Sphingobacterium*, *Ochrobacterium*, *Stenotrophomonas*, *Shinella*, *Diaphorobacter*, *Agrobacterium*, *Achromobacter*, *Phenylobacterium*, and *Flavobacterium* [5]. This differentiation of sewer bacterial WW communities was further confirmed after a passage through a wastewater treatment plant (WWTP) [6]. Urban- and hospital-treated WW showed distinct bacterial communities matching their origin but also their chemical contents [6]. The hospital WW (HWW) was found to contain specific pharmaceuticals not found in a domestic sewer such as iohexol, amoxicillin, and flucloxacillin [7]. The domestic WW showed higher occurrences of pesticides such as organochlorines, and of motor engine contaminants such as hydrocarbons, or metallic trace elements [8] like lead, copper, and cadmium [9,10,11]. 

A selective effect, even at low concentrations, of these WW chemical pollutants on sewer microbial communities was suggested to occur. Antibiotics were found to modulate WW microbial communities by driving a decay of sensitive bacteria and selecting resistant species [12,13]. Furthermore, the greater uses of antibiotics in hospital settings were considered to have increased the occurrences of multi-resistant bacteria among the receiving sewer networks [14]. Positive correlations between concentrations of sulfamethoxazole, tetracycline, and ciprofloxacin in WW and counts of bacteria of health concern such as respectively *Acinetobacter*, *Clostridia*, *Aeromonas,* and *Alistipes* were reported [15]. Furthermore, antibiotic resistance genes (ARGs) were found in greater numbers in hospital sewers than domestic ones through qPCR assays for *bla*_TEM_ (penicillin resistance), *qnrS* (fluoroquinolone resistance), *sulI* (sulfonamides resistance), *mecA* (methicillin-resistance gene), and *tetW* and *tetO* (tetracycline resistance) [16,17]. These greater numbers of ARG could be related to their occurrences among mobile genetic elements (MGEs) such as plasmids, integrons, and transposons which could have spread in the bacterial communities [18]. Poor decay rates of the hospital (H) WW chemical and biological contaminants thus represent a significant public health concern [19]. 

The main goal of this study was to assess the ability of WWTP-related bacteria at competing with native freshwater microbial communities and at occupying particular river ecological niches. An experimental framework was designed to demonstrate that HTWW (hospital-treated WW) and DTWW (domestic-treated WW) bacteria released into a stream not only can get mixed with endogenous river taxa but can also interact with the native communities to favor the formation of epilithic biofilms. Rock surfaces were considered growth opportunities for WWTP-related taxa with strong biofilm development capacity and high tolerance to chemical pollutants which are simultaneously delivered with the TWW (treated wastewaters). To test these hypotheses, blueschist rocks were exposed over 8 to 16 weeks to river surface waters and treated wastewater, and their biofilm bacterial communities and pharmaceutical contents were analyzed and compared. These investigations were performed using the French Sipibel experimental site [20].

The rock bacterial biofilm community structures were investigated by classical most probable number growth assays and platings, by quantitative PCR assays targeting DNA signatures of integrons and fecal source indicator species, and through the *tpm* metabarcoding approach. This metabarcoding approach was found to be highly discriminant and to allow the differentiation of more than 100 species of γ-proteobacteria such as those of the *Aeromonas* and *Pseudomonas* which are highly prevalent in WW [21]. The contingency table of amplicon sequence variants (ASV) of these *tpm*-harboring taxa could be used in a SourceTracker Bayesian approach [22] to infer contributions of WWTP-related taxa in the build-up of river epilithic biofilms. This *tpm* source tracking revealed a significant number of sewer *tpm* ASV among river epilithic biofilms, even at a distance of up to 8 km from the WWTP outlet. These observations demonstrated the ability of WWTP *tpm*-harboring bacterial taxa at invading major river ecological niches and competing with native freshwater taxa.

## 2. Materials and Methods

### 2.1. Study Site

The WWTP of Bellecombe (http://www.graie.org/Sipibel/ (accessed on 16 February 2023)) was used in this study. It has a processing capacity of 32,000 equivalent inhabitants. The outflows of this WWTP are delivered into the Arve river which flows through Switzerland and France over 108 km before reaching the Rhône river. Flow monitoring indicated an average of 66 m^3^ of water discharged per second close to the WWTP with a maximum flow in May and June. The Bellecombe WWTP processes are described in Chonova et al. [6] and Wiest et al. [23]. Two treatment files were implemented at this site, one for the domestic WW (DWW) (20,850 inhabitants) and one for the hospital WW (HWW) from the Alpes Léman hospital (CHAL) (450 beds). 

### 2.2. Sampling Strategy

The bacterial community studies were based on the analysis of biofilms that had developed on blueschist rocks exposed to DTWW (domestic-treated WW), HTWW (hospital-treated WW), or the Arve surface waters. This approach allowed an investigation of the co-accumulation of micro-organisms and chemical pollutants in the collected biofilms, and to highlight co-occurrence patterns between these contaminants. Before being used on site, blueschist rocks were brushed, cleaned with 5% hyperchlorite, and sterilized by autoclave. Blueschist rocks were placed in stainless steel traps which were immerged under 5 cm of water in the various compartments to be tested: (1) in the clarifiers of the hospital (b-HTWW; n = 6 samples) and domestic (b-DTWW; n = 6) lines, and (2) in the Arve river at 8 km upstream and downstream the entry point of the treated effluents (b-RU_far and b-RD_far: 8 km; n = 6 for each), at 500 m upstream and downstream the entry point (b-RU_med and b-RD_med; n = 6 for each) and at 10 m upstream and downstream the entry point (b-RU_close and b-RD_close; n = 5 for each) (see Figure 1). Sampling dates are indicated in Appendix A. Six sampling campaigns were carried out except for b-RD_med and b-RU_med for which only five could be conducted because of bad field conditions. Blueschist rocks were immerged for two to four months, depending on the biofilm growth rates. At the same time as biofilm sampling, water was also collected from the same compartments and was analyzed in terms of pharmaceutical contents and basic physicochemical parameters (Appendix A). 

### 2.3. Physico-Chemichal Analyses

Physicochemical parameters of blueschist rock biofilms or water samples were monitored (Appendix A). Nutrients (NO2-NO3-PO4), ammonium, phosphate, TSS, and DOC (dissolved organic carbon) were measured according to the French standard operating procedures (AFNOR 1997 [24]) for water. Moreover, 18 pharmaceuticals divided into NSAIDs (non-steroidal anti-inflammatory drugs), beta-blockers, antibiotics, anticonvulsants, and analgesics were quantified in the surface waters and the sampled biofilms according to Chonova et al. [6]: Aciclovir, Atenolol, Bezafibrate Caffeine, Carbamazepine, Clofibrate, Diazepam, Diclofenac, Epoxy-carbamazepine, Iohexol Ketoprofene, Levofloxacin, Metronidazole, Propranolol, Ranitidine, Roxithromycine, Sulfamethoxazole, Trimethoprim. Rain precipitation regimes were monitored by Météo-France at Contamine-sur-Arve (France) which is 8 km away from the Bellecombe WWTP. Solar irradiance was measured by the INRAE meteorological station. Arve river temperatures were monitored at the station of the Federal Office for the Environment of Switzerland (Geneva, Switzerland). WW retention time per clarifier and effluent discharge volumes were measured by the SIPIBEL observatory [20]. 

### 2.4. Bacterial Counts and DNA Extractions

Counts of total heterotrophic bacteria were estimated by inoculation of 10-fold diluted TSA (tryptic soy agar) amended with 1.2% agar. *Aeromonas caviae* and *Pseudomonas aeruginosa* were quantified using the MPN (most probable number) methods described in Navratil et al. [25]. Intestinal *Enterococci*, total coliforms, and *E. coli* were measured using the IDEXX Colilert and the Enterolert type E kit (IDEXX, Westbrook, NC, USA) following the manufacturer’s instructions. Biofilms on the immerged Blueschist rocks were scraped using a sterile spatula and a sterile toothbrush. Microbial DNAs of biofilms were extracted with the Gene-Elute-LPA kit (Sigma-Alrich, Saint-Louis, MO, USA) using the manufacturer’s instructions. DNA extracts were used in qPCR assays performed in triplicates on a Bio-Rad CFX96 qPCR apparatus using the Brilliant II SYBR Green low ROX qPCR mix (Agilent, Vénissieux, France). Quantifications of the gene targets were performed using the CFX Manager 3.0 software (Bio-Rad, Marnes-la-Coquette, France). Total bacteria were quantified by qPCR using the 338F and 518R primers targeting the 16S rRNA gene according to [26]. Total *Bacteroidales* were quantified using the AllBac296F and AllBac467R primers according to [27]. Relative contents in Human fecal bacteria of the collected samples were inferred using the HF183 qPCR assay [28]. Mobile genetic elements (MGEs) named integrons were quantified by targeting integrase genes of class I, II, and III according to [29]. 

### 2.5. The tpm Metabarcoding Analytical Scheme 

The *tpm* phylogenetic marker allows an allocation of amplicon gene sequences to bacterial species of, mainly, a group of γ-proteobacteria including the *Aeromonas* and *Pseudomonas*. PCR products were generated as already described [21]. PCR products were purified and sequenced by Biofidal (Microsynth, Vaux-en-Velin, France). Raw reads are available on the European Nucleotide Archive with the accession number PRJEB36958. *tpm* raw reads were processed using the dada2 package for R [30]. Linker, barcode, and primers were removed using the Trimgalore v0.6.5 software (https://www.bioinformatics.babraham.ac.uk/projects/trim_galore/ (accessed on 16 February 2023)). The SOP (standard operating procedure) at https://benjjneb.github.io/dada2/bigdata_paired.html (accessed on 16 February 2023) for paired-end reads was used to process the *tpm* raw reads. The decontam package for R [31] was used to detect contaminant amplicon sequence variants (ASV) from control samples. Only 21 contaminant ASV representing 0.27% of the total reads were detected. The *tpm* reads were taxonomically affiliated using the *tpm* metabarcoding sequences database [32]. 

### 2.6. Statistical Analyses 

Alpha (Shannon, Simpson, and Evenness) and beta diversity (Bray-Curtis dissimilarity distances) indices and rarefaction curves were computed using the R software v3.5.3 with the Vegan package v2.5.6 as carried out in [33]. All Wilcoxon tests were also computed using the R software. Distribution biases of the *tpm* inferred taxa between the b-HTWW, the b-DTWW, and the river biofilms samples were investigated using the DESeq2 package for R [34]. Spearman correlation matrices between the inferred *tpm* species and the concentration of pharmaceuticals were computed using the Corrplot R package v0.84. Spearman correlation matrices were illustrated by a heatmap computed on the Excel software. Coalescence phenomena between bacterial communities of the treated wastewater effluents (contamination sources) and the river ones (receiving environment) were investigated using the SourceTracker computer package described by [22]. The Bayesian SourceTracker analyses were computed three times with the default parameters (rarefaction depth  = 1000 reads from the original cleaned dataset of *tpm* gene read, burn-in: 100, restart: 10). The confidence on the computed contribution values were estimated using the relative standard deviation (RSD) from the three SourceTracker runs (e.g., [35]).

### 2.7. Pseudomonas aeruginosa Isolation and Antibiotic Resistance Assays 

*P. aeruginosa* strains were isolated from HTWW, DTWW, and river surface waters but also from the raw WW from the hospital and domestic sewers. *P. aeruginosa* strains were isolated using a *Pseudomonas* agar base (Oxoid, Basingstoke Hampshire, UK) completed with 5% glycerol, 200 mg/L cetrimide, and 15 mg/L nalidixic acid. Isolates were confirmed as belonging to *P. aeruginosa* using the *ecfX*-specific PCR screening [36]. Antibiograms were carried out following the recommendations of the French Society of Microbiology (SFM) for investigations of *P. aeruginosa* antibiotic resistance using n = 188 Sipibel field site isolates. 

## 3. Results

### 3.1. Monitoring of Pharmaceutical Loads, Physico-chemical, and Classical Microbial Parameters

Pharmaceuticals, physicochemical, and classical bacterial parameters were monitored for the rock biofilms and water samples from the river, and domestic and hospital treatment lines of the WWTP of Bellecombe (Appendix A). Principal component analyses (PCA) were performed on these datasets (Figure 2 and Appendix A). These PCA clearly segregated b-HTWW and b-DTWW rock biofilms (b) according to their contents in pharmaceuticals (Figure 2). Atenolol, diclofenac, propranolol, and trimethoprim concentrations were higher in the b-DTWW biofilms than the hospital ones (termed b-HTWW). Wilcoxon tests confirmed these trends with significantly higher concentrations of atenolol in the b-DTWW (*p*-value < 0.05). Iohexol, ranitidine, levofloxacin, and roxithromycin were found in significantly higher concentrations in b-HTWW than in b-DTWW (Figure 2) (Wilcoxon test: *p*-value < 0.05). Most of these trends were confirmed with the WW monitoring (Appendix A). In fact, carbamazepine, aciclovir, epoxy-carbamazepine, diazepam, metronidazole, and iohexol concentrations were significantly higher among the HTWW (Wilcoxon test: *p*-value < 0.05) (Appendix A). Atenolol, ketoprofen, diclofenac, bezafibrate, and trimethoprim were in higher concentrations in DTWW (Wilcoxon test: *p*-value < 0.05). Correlograms computed from Spearman correlation matrices between the pharmaceutical concentrations monitored in biofilms or WW confirmed these segregations between the domestic and hospital treatment lines (Appendix A). It is to be noted that pharmaceuticals could not be quantified among the river biofilms because of contents below the levels of detection (Appendix A). Higher concentrations of phosphate in the b-HTWW and ammonium in the b-DTWW were observed (Wilcoxon test: *p*-value < 0.05). Interestingly, higher caffeine contents were observed among the DTWW than HTWW (Wilcoxon test: *p*-value < 0.05) but no major differences were observed between the b-HTWW and h-DTWW (Wilcoxon test: *p*-value > 0.05). 

Microbial contents of the biofilm samples were monitored by culture- and DNA-based methods including quantitative PCR assays. This highlighted high 16S rRNA gene copy numbers from 1.41 × 10^5^ to 1.8 × 10^7^ per g biofilm (dry weight) (Appendix A). River biofilms and b-HTWW/DTWW did not show significant differences in these copy numbers over time (Wilcoxon tests; *p*-value > 0.05). Total *Bacteroidales* per g biofilm ranged from 0 to 1.4 × 10^3^ copies per g biofilm (dry weight). Generally, fecal indicator bacteria (intestinal *enterococci*, *E. coli,* and total coliforms) were found in higher numbers among the b-DTWW samples than b-HTWW (Wilcoxon-test; *p*-value < 0.05). A PCA was computed from this dataset (Appendix A). b-DTWW samples were characterized by higher counts of class I and III integrons (Wilcoxon test; *p*-value < 0.05). Interestingly, b-RU_far5 (8 km upstream of the WWTP) river biofilms showed higher numbers of integron II DNA sequences (Appendix A). Most microbial parameters of the river biofilms and b-HTWW samples could not be differentiated by PCA. *P. aeruginosa* and *A. caviae* MPN counts showed homogeneous distribution patterns over the study among the b-HTWW and h-DTWW (Wilcoxon-test; *p*-value > 0.05). Nevertheless, high numbers of *P. aeruginosa* gene targets of 3.67 × 10^1^ to 8.05 × 10^4^ and 6.83 × 10^3^ to 7.04 × 10^6^ MPN/g biofilm were detected among, respectively, the b-HTWW and b-DTWW. *P. aeruginosa* numbers were below the limit of detection for the biofilms sampled from the Arve River. 

Relationships between the above bacterial datasets and pharmaceutical concentrations recorded among biofilm samples were explored by Spearman correlation tests (Appendix A). All bacterial counts except the ones of the total heterotrophic bacteria and integron III were found to be positively correlated to concentrations of drugs most often affiliated to the b-DTWW treatment line (e.g., atenolol, diclofenac, propanolol) and negatively to the b-HTWW samples (Figure 2 and Appendix A). Integron III numbers showed positive correlations with drug concentrations most often associated with the b-HTWW (iohexol, levofloxacin, ranitidine, roxithromycin). *P. aeruginosa* numbers were found to be positively correlated to pharmaceutical loads of both the b-HTWW and b-DTWW and concentrations of aciclovir, atenolol, clofibrate, diazepam, and epoxicarbamazepine (Appendix A). *P. aeruginosa* numbers were negatively correlated to roxithromycin concentrations. Roxithromycin was previously shown to inhibit biofilm formation by *P. aeruginosa* when associated with other antibiotics [37].

### 3.2. Genetic Structure of tpm Bacterial Communities among Biofilms

#### 3.2.1. General Features

The *tpm* Illumina PCR primers defined in Aigle et al. [21] generated PCR products from all DNA samples, and these could be sequenced by the Illumina MiSeq v3 technology. Processing of the sequenced PCR products generated a dataset of 1 034 771 sequences distributed into 3082 ASV (Appendix A). Biofilm river samples b-RD_far6, b-RD_med6, and b-RU_med_6 showed the highest ASV numbers with respectively 826, 807, and 730 ASV (Appendix A). Conversely, biofilm samples b-HTWW3 (124 ASV), b-HTWW1 (161 ASV), and b-HTWW4 (180 ASV) showed the lowest ASV numbers (Appendix A; Appendix A). Biofilms sampled from river Blueschist rocks located upstream (b-RU) or downstream (b-RD) of the WWTP outlet showed significantly higher ASV numbers than those of the b-HTWW and b-DTWW (Wilcoxon test; *p*-value < 0.001). There was no significant difference between ASV numbers of biofilms from b-RU and b-RD samples. However, the distance between sampling points and the outlet of the WWTP impacted negatively the ASV numbers. B-RU_close and b-RD_close biofilms showed significantly lower numbers of ASV than those sampled from b-RU_med, b-RU_far, b-RD_med, and b-RD_far (Wilcoxon tests; *p*-value < 0.01). b-HTWW biofilms showed significantly lower ASV numbers than those from the b-DTWW ones (Wilcoxon test; *p*-value < 0.05). Shannon index computed from the *tpm* ASV datasets showed heterogeneous values going from 5.14 to 1.11 (Appendix A). Sample b-RD_med4, b-RD_med1, and b-RD_med5 showed the highest Shannon values with respectively 5.14, 5.12, and 5.02 while b-RU_far3, b-RU_far4, and b-HTWW6 showed the lowest ones with 1.11, 2.75 and 3.10. b-DTWW biofilms showed significantly lower Shannon indices than those of the river and the hospital line (Wilcoxon test; *p*-value < 0.01). Evenness indices showed values from 0.98 to 0.17. b-HTWW3 (0.98), b-HTWW4 (0.89) and b-RU_close3 (0.86) showed the highest values while b-RU_far3 (0.17), b-RU_far4 (0.45) and b-RD_far1 (0.50) showed the lowest ones (Appendix A). Distance between biofilm sampling points and the outlet of the WWTP did not significantly change the Shannon and Evenness indices (Wilcoxon test; *p*-value > 0.05). 

In order to compare *tpm* bacterial communities of the biofilm samples according to their origin, a Bray-Curtis dissimilarity matrix was computed on the ASV table after a Hellinger transformation (Appendix A). NMDS ordinations were computed from the Bray-Curtis dissimilarity distances (Figure 3). NMDS clearly separated b-HTWW, b-DTWW, and river biofilm samples. These observations were supported by significant ANOSIM tests (*p*-value < 0.001). However, ANOSIM tests did not show significant differences between river biofilms sampled upstream and downstream of the outlet of the WWTP (*p*-value > 0.05). Taxonomic affiliations of the ASV were performed. This allowed the classification of more than 99% of the reads at the phylum level (Appendix A). As expected, *Proteobacteria* were dominant in the *tpm* dataset and represented 96.4% of the tracked taxa. The *tpm* marker also allowed the detection of *Bacteroidetes* (0.01%) and *Cyanobacteria* but in low numbers. At the genus level, taxonomic affiliations allowed the classification of about 80% of the *tpm* reads into 24 well-defined genera (Appendix A). 

ASV were mainly allocated to the *Pseudomonas*, *Hyphomicrobium*, *Aeromonas, and Herbaspirillum* at respectively 41, 28, 4.5, and 2.4% relative abundances (Appendix A). Interestingly, *Aeromonas* ASV showed relative abundances higher among biofilms sampled from less impacted compartments (b-RU_med, b-RU_far, b-RD_med, b-RD_far; Wilcoxon test; *p*-value < 0.05) than the b-TWW related ones. Nevertheless, *Aeromonas*, *Pseudomonas,* and *Herbaspirillum* were found among all the biofilm samples (Appendix A). Taxonomic affiliations of *tpm* ASV at the species level allowed the tracking of 91 well-defined species representing about 25% of the *tpm* harboring bacteria resolved by the metabarcoding approach deployed in this work (Appendix A). Five species dominated the *tpm* dataset with relative abundances higher than 1%: *P. anguilliseptica* (6.85%)*, P. anguilliseptica-like* (4.80%), *H. aquaticum* (2.41%), *A. sobria* (1.45%), *P. fluorescens* (1.19%) and *P. koreensis* (1.14%). Several species of health concern were detected among the *tpm* dataset including *A. sobria*, *P. aeruginosa*, *A. caviae*, and *A. hydrophila*. Infra-specific diversity of the *P. aeruginosa* reads was explored according to Aigle et al. (2021). ASV_1211 and ASV_1373 were found to match the *tpm* type-G08-24 of the *P. aeruginosa* PA14 sub-clade. ASV_1664 and ASV_1682 matched the *tpm* type-G12_15_17_18_35 sequences of the PAO1 sub-clade. Two fish pathogens, *P. anguilliseptica* and *A. salmonicida,* could be tracked using the *tpm* dataset.

#### 3.2.2. *tpm* ASV Specific of Biofilms Generated from DTWW and/or HTWW

Distribution biases of the *tpm* ASV allocated to particular species were investigated (Appendix A). Eleven well-defined species were detected among the b-TWW (meaning both HTWW/DTWW) biofilms including *Lamprocystis purpurea* (5/12 samples) and *Stenotrophomonas humi* (4/12) (Appendix A). *Nitrospira defluvii tpm* ASV were present in most of the b-TWW biofilm samples and were less common among the river ones (b-TWW: 12/12 samples; b-RU: 6/17) (Appendix A). DESeq2 analyses confirmed these trends and highlighted other species showing higher read numbers among the b-TWW biofilm samples than the river ones. The *P. anguilliseptica-like*, *H. aquaticum*, *P. brenneri,* and *P. pohangensis-like* ASV reads showed higher numbers among b-TWW than river biofilm samples (DESeq2: *p* < 0.05; Appendix A). Reads allocated to *P. fluorescens*, *P. grimontii*, *A. sobria,* and *A. eucrenophila* were found to be higher among the river biofilm samples (DESeq2: *p* < 0.05; Appendix A) than the b-TWW ones. DESeq2 analyses also allowed exploring distribution biases between the DTWW and HTWW treatment lines (Appendix A). Read numbers affiliated to *L. purpurea* were higher among the b-DTWW samples (DESeq2: *p* < 0.05; Appendix A). *S. humi* and *A. popoffii* showed higher read numbers among the b-HTWW (DESeq2: *p* < 0.05). No species were found to vary significantly between b-RU and b-RD river biofilm samples (DESeq2: *p* > 0.05). 

DESeq2 computations also highlighted significant distribution biases at the sub-species level by considering variations per ASV (Appendix A). In fact, 67 *tpm* ASV including highly prevalent ones showed differentiated patterns according to their occurrences among b-HTWW or b-DTWW biofilms (Appendix A). To illustrate, ASV_6 (n = 27,917 reads; unclassified *Gammaproteobacteria*), ASV_42 (n = 3622 reads; unclassified bacteria), ASV_52 (n = 2789 reads; unclassified *Proteobacteria*) or ASV_131 (n = 1024 reads; unclassified *Proteobacteria*) showed higher read numbers among b-HTWW samples (DESeq2: *p* < 0.05). Conversely, ASV_22 (n = 7048 reads, *Pseudomonas* sp. PGPPP3), ASV_39 (n = 3945 reads, *Pseudomonas* sp. PGPPP3), ASV_59 (n = 2267 reads; unclassified *Pseudomonas*) and ASV_84 (n = 1849 reads; *P. anguilliseptica-like*) were found in higher numbers among the b-DTWW samples (Appendix A). Additionally, more than 175 ASV showed significant distribution biases between biofilm samples exposed to HTWW or river surface waters (Appendix A). This included highly prevalent ASV such as ASV_1 (n = 76197 reads; *Hyphomicrobium* sp.), ASV_2 (n = 49,970 reads; *Hyphomicrobium* sp.), and ASV_63 (n = 2196 reads; *P. grimontii*) which showed higher read numbers among river biofilms (DESeq2: *p* < 0.05). Conversely, nine ASV were found in higher numbers among biofilms exposed to TWW: ASV_46 (n = 3173 reads; *Dechloromonas* sp.), ASV_93 (n = 1573 reads; *N. defluvii*), ASV_96 (n = 1490 reads; *P. anguilliseptica-like*) and ASV_148 (n = 964 reads; *N. defluvii*) (DESeq2: *p* < 0.05) (Appendix A). *Dechloromonas* and *Nitrospira* spp. are, respectively, the dominant denitrifiers and nitrite-oxidizing bacteria of wastewater treatment plants. Only ASV_47 (n = 3028 reads; *A. sobria*) showed higher read numbers among biofilms sampled upstream of the WWTP rather than downstream (DESeq2: *p* < 0.05) (Appendix A).

#### 3.2.3. Relationships between the Relative Content of *tpm*-Harboring Species and Pharmaceuticals among b-HTWW and b-DTWW

A heat map illustrating relations between the *tpm* metabarcoding dataset and the pharmaceutical concentrations was computed and highlighted significant Spearman correlations (Appendix A). Among the most significant correlations (r > 0.8, r < −0.8), reads allocated to *L. purpurea* were found positively correlated to the concentrations of atenolol (D) (D indicates higher concentrations in the domestic line biofilms, and H in the hospital one according to Figure 2), propranolol (D) and diclofenac (D) (Figure 4), and those allocated to *Pseudomonas*_Gp_BR were correlated to caffeine concentrations. However, *A. popoffii* and *S. humi tpm* reads were negatively correlated to atenolol (D), those of *Pseudomonas*_Gp_BR to bezafibrate, of *A. veronii* to caffeine, and of *L. purpurea* to iohexol (H) and ranitidine (H). *L. purpurea tpm* reads also showed significant positive correlations with concentrations of trimethoprim (D), and negative ones with iohexol (H), ranitidine (H), roxithromycin (H), and sulfamethoxazole (H) as indicated in Appendix A and Figure 4. *S. humi* and *A. popoffii* read numbers were positively correlated to iohexol (H), levofloxacin (H), ranitidine (H), roxithromycin (H), and sulfamethoxazole (H), but negatively to clofibrate, proponolol (D) and trimethoprim (D) (Appendix A, Figure 4). The Human opportunistic pathogen *P. aeruginosa* read numbers were found to be positively correlated with aciclovir, atenolol (D), and bezafibrate, but negatively with roxithromycin (H) concentrations (Appendix A). Among the most frequent taxa, *P. anguilliseptica* was found to be negatively correlated to concentrations of bezafibrate, levofloxacin (H), and roxithromycin (H), and positively with caffeine. *P. anguilliseptica-like tpm* reads showed negative relations with trimethoprim, which was in higher concentrations in the b-DTWW but also clofibrate, and diazepam. *H. aquaticum tpm* read numbers were found correlated to bezafibrate and trimethoprim (D), and negatively with diazepam and levofloxacin (H). 

### 3.3. Coalescence of b-DTWW and b-HTWW tpm Bacterial Communities among Epilithic Biofilms 

Coalescence of the *tpm* ASV from the b-HTWW and b-DTWW (termed b-TWW when grouped) biofilms with those observed in epilithic biofilms upstream of the WWTP outlet was estimated among downstream WWTP outlet epilithic biofilms by a SourceTracker Bayesian analysis (Table 1). These analyses revealed significant coalescence between the “source” bacterial communities among the WWTP-impacted epilithic biofilms (considered the sink). The *tpm* bacterial taxa of the b-DTWW biofilms were found to explain on average 8.8% of the diversity found among downstream WWTP river biofilms. RD_close river biofilm samples were significantly more impacted by b-DTWW *tpm* bacterial communities than biofilms at b-RD_med and b-RD_far sampling sites (Wilcoxon-test; *p*-value < 0.05). *tpm* bacterial communities of downstream WWTP biofilm river samples were also found to be explained by about 0.7% of the b-HTWW biofilm ASV (Appendix A). Logically, b-RD_med and b-RD_far river biofilms showed higher numbers of ASV from the river biofilms sampled upstream of the WWTP (respectively 61.62 and 62.23%) than those of b-RD_close (28.14% of contribution; Wilcoxon-test; *p*-value < 0.05) (Appendix A). On the opposite, b-RD_close biofilm samples showed the highest proportion of unknown sources of recovered *tpm* sequences (28.14% of contribution; Wilcoxon-test; *p*-value < 0.05). These trackings of *tpm* bacterial sequences going from b-TWW towards river biofilms were further explored using a Venn diagram illustrating the origin of the ASV (Figure 5). Biofilm samples from all compartments (b-HTWW, b-DTWW, b-RU, and b-RD) shared 161 ASV (Figure 5). b-DTWW, b-HTWW, b-RU, and b-RD harbored respectively 250, 225, 341, and 449 specific ASV. A total of 55 ASV were found to be specific to b-TWW (b-DTWW and b-HTWW) and 1243 were specific to the river biofilms (b-RU and b-RD) (Figure 5). b-RU shared only 8 ASV with b-TWW. Coalescence phenomena were confirmed with the detection of 15 ASV shared between b-TWW and b-RD including ASV_96 (n = 1490 reads; *P. anguilliseptica-like*), ASV_186 (n = 700 reads; *P. extremaustralis*), ASV_402 (n = 290 reads; *Nitrosomonas* sp.), ASV_928 (n = 118 reads; *P. parafulva*) and AVS_1173 (n = 93 reads; *P. pohangensis-like* (Appendix A). 

### 3.4. Antibiotic Resistance of P. aeruginosa Strains

*P. aeruginosa* was detected among all samples either by PCR assays or the *tpm* metabarcoding approach, this species was further used to explore the transfer of antibiotic-resistant strains from the hospital and domestic sewer systems down into the connected river. *P. aeruginosa* strains were isolated from wastewater of the clarifiers of the hospital and domestic sewer lines of the WWTP, and the Arve River. Antibiograms were carried out on 188 isolates (DWW: 40; HWW: 44; b-DTWW: 46; b-HTWW: 52; b-RU: 1; b-RD: 5 strains) (Appendix A). The *P. aeruginosa* isolates of the hospital sewer line showed the broadest range of antibiotic resistances (up to 15 distinct resistances). Over this collection, 57 strains were found to be multi-resistant (showing resistances to at least three antibiotic families) and 40% of these had been isolated from the HWW treatment line (Appendix A). These isolates showed resistance to beta-lactamines, aminosides, and sulfamides, and some from HTWW were also resistant to fluoroquinolones (Appendix A). The isolation of resistant strains from HTWW indicated a poor decay rate of *P. aeruginosa* among the WWTP. Among the five *P. aeruginosa* isolates obtained from the river samples collected downstream the WWTP overflow, one was found multi-resistant (bpoe4818, resistant to ticarcillin, ticarcillin associated to clavulanic acid, ertapenem, norfloxacin and trimethoprim associated with sulfamethoxazole) and likely to come from the hospital WW. Several strains of the hospital line were found highly resistant to the tested antibiotics such as bpoe3519, bpoe3454, or bpoe3516 isolated from the HWW samples and showing respectively 15, 14, and 14 antibiotic resistances over the 21 tested conditions (Appendix A). Strains from HTWW also showed high numbers of antibiotic resistance. Multi-resistant strains from the domestic line (DTWW) harbored not more than five resistances. Nevertheless, bpoe4092 isolated from DTWW was resistant to colistin which is the ultimate last solution antibiotic to treat *P. aeruginosa* infections (Appendix A). It is to be noted that the resistance patterns did not match the recorded antibiotics monitored in the (b-) HTWW or (b-) DTWW. 

## 4. Discussion

Chemical and biological contaminants of sewer systems can greatly vary according to the origin of the wastewater [38]. At the Sipibel experimental site used in this study, hospital and domestic WW could be treated by distinct WWTP facilities prior to their release into a connected river. This allowed building up an experimental strategy making possible tracking of the contaminants from each sewer line into the connected river. These trackings were performed through the analysis of biofilms accumulating over blueschist rocks while being incubated in the treated wastewater of the WWTP clarifiers of the experimental site, and in the river surface waters downstream of the WWTP outlet. The chemical and microbiological contents of these biofilms were expected to be representative of the main pharmaceuticals and microbial taxa likely impacting the ecological quality of the epilithic biofilms developing in the connected river. The main conclusions of these analyses are shown in Figure 4. 

These investigations showed atenolol, diclofenac, propranolol, and trimethoprim concentrations to be higher in the biofilms of the domestic line (b-DTWW) of the Sipibel WWTP rather than of the hospital one. However, iohexol, ranitidine, levofloxacin, roxithromycin, and ketoprofen were found in significantly higher concentrations in the biofilms developing from the treated hospital WW. Furthermore, more antibiotics were globally found in the b-HTWW than in the b-DTWW. A *tpm*-metabarcoding approach was then applied to explore the co-occurrence patterns of pharmaceuticals and bacterial species from over 20 genera including *Pseudomonas* among these biofilms [21]. The river biofilms showed the highest numbers of *tpm* ASV (up to 800) recovered in this study suggesting a strong ability of the *tpm*-harboring bacterial taxa at getting established in a river system. The treated WW (TWW) biofilms were made of about 100 to 300 ASV, and the 16S rRNA gene qPCR assays did not show significant differences in terms of total bacterial counts per investigated biofilm whatever their location and sampling period. Their contents varied between 1.41 × 10^5^ to 1.8 × 10^7^ 16S rRNA gene copy numbers per g of biofilm (dry weight). Most of these biofilms were found to contain the HF183 marker indicative of the occurrences of significant exposures to Human fecal bacterial releases, and also of the DNA markers indicative of the occurrences of integrons that can spread ARGs. A comparison of these *tpm* ASV distribution patterns between samples was performed. NMDS ordinations of Bray-Curtis dissimilarities indicated significant differences between biofilm community patterns of the hospital (b-HTWW) and domestic lines (b-DTWW) and of the connected river. Previous analyses using denaturing gradient gel-electrophoresis (DGGE) also indicated such a differentiation of the hospital (b-HTWW) and domestic lines (b-DTWW) bacterial communities but suggested a lower diversity in the b-HTWW [6]. The *tpm*-metabarcoding approach implemented here rather indicated significantly lower Shannon indices for *tpm*-harboring bacteria among the domestic (b-DTWW) than the hospital (b-HTWW) and river biofilms. 

Among the *tpm*-harboring bacterial cells growing in the biofilms, 15 ASV (allocated respectively to a *P. anguilliseptica*-like species, *P. extremaustralis*, *Nitrosomonas* sp., *P. parafulva*, and a *P. pohangensis-like* species) were found to be shared between the b-HTWW, b-DTWW and the river biofilms downstream (b-RD) the WWTP outflow. Otherwise, 13 ASV were strictly shared between the b-HTWW and b-RD (river biofilm downstream the WWTP) biofilms, and 40 between the b-DTWW and b-RD biofilms. In addition to these strict presence/absence distribution patterns, up to 67 *tpm* ASV were found to vary significantly between the b-HTWW and b-DTWW biofilms. Some of these showed very high fold changes between the two sewer lines such as ASV_6 (an unclassified *Gammaproteobacteria*) (but also more than 20 additional ASV) that was found in much higher numbers in the hospital line biofilms (b-HTWW). On the opposite, ASV_84 (a *P. anguilliseptica-like* species) and more than 40 additional ASV were in significantly higher relative numbers in the domestic WW-related biofilms (b-DTWW). In order to further explain these differences, co-occurrences between the *tpm*-harboring species and pharmaceuticals were further investigated. 

Several positive correlations between the *tpm* metabarcoding libraries and pharmaceutical loads of the biofilms were observed. The phototrophic purple sulfur bacterium growing in the biofilms named *L. purpurea* was found to be the main *tpm*-harboring species showing distribution biases matching pharmaceutical concentrations. This species can use sulfur as the electron donor of its photosynthetic pathway [39]. Here, its distribution pattern was positively correlated with the concentrations of atenolol, propranolol, and diclofenac which were found higher in the domestic sewer biofilms. On the opposite, two other highly prevalent *tpm*-harboring species, *S. humi,* and *A. popoffii,* showed *tpm* read number distribution patterns rather indicative of a tropism for the hospital sewer system. The *S. humi* and *A. popoffii tpm* read numbers were found positively correlated to the higher concentrations of iohexol, levofloxacin, ranitidine, roxithromycin, and sulfamethoxazole found in the hospital biofilms. This could be related to the pathogenic nature of *A. popoffii*, and its implication in urinary tract infections [40]. *A. popoffii* could thus be a potential indicator of hospital releases. The association of *S. humi* with the hospital sewer network could be related to its involvement in denitrification [41]. Further analyses will be required to explore its relative contribution in a WWTP nitrogen transformation processes. The *tpm* sequences of other Human opportunistic pathogens such as *A. caviae*, *A. sobria*, *P. aeruginosa*, *P. flavescens,* and *P. putida* were detected in this study, but their distribution patterns were not highly correlated to the monitored pharmaceuticals (Spearman correlation rho factors < 0.8). Nevertheless, *P. aeruginosa* read numbers were found to be negatively correlated to roxithromycin found in the hospital b-HTWW biofilms. This latter observation was further supported by the MPN growth assays. This is in line with the previous report showing roxithromycin to inhibit biofilm formation by *P. aeruginosa* when associated with other antibiotics [37]. Interestingly, this distribution pattern was found associated with a higher recovery from the hospital sewer network of *P. aeruginosa* isolates showing multiple antibiotic resistances (up to 15 antibiotic resistances). 

Contributions of the WWTP *tpm*-bacterial taxa coming from the hospital and domestic lines in the build-up of the river biofilm bacterial communities were further investigated using a Bayesian source-sink tracking approach [22]. This SourceTracker analysis allowed us to show a significant coalescence of WWTP *tpm*-harboring taxa among the river epilithic biofilms. Biofilms recovered at 10 m from the WWTP outlet showed up to 35% of their *tpm* ASV contents to have originated from the domestic sewer line (b-DTWW). Furthermore, hospital-borne *tpm* ASV (b-HTWW) were found to have contributed to the build-up of these communities. These b-HTWW taxa represented up to 2.5% of the epilithic biofilm *tpm* bacterial communities. Such signatures of sewer *tpm* ASV could even be recorded among the epilithic biofilms collected at 8 km from the WWTP outlet, and at percentages varying between 0.16 to 23% of the total *tpm* community. The observed differences in contributions of *tpm*-bacterial taxa between the sewer lines were likely related to the higher bacterial discharges coming from the domestic WWTP than the hospital one. The average discharge volumes coming from the DTWW network were estimated at 2198 m^3^/day while, from the HTWW one, they were estimated at 140 m^3^/day [20]. Interestingly, it is to be noted that the Arve river at the sampling points showed a day dilution of these discharges by over 5 million m^3^ of surface waters (See http://valleedelarve.n2000.fr/la-plaine-alluviale-de-l-arve/le-bassin-versant-de-l-arve/la-riviere-arve (accessed on 16 February 2023)). This indicates that, even if the Arve river can significantly dilute these exogenous taxa, some were still highly active and could efficiently colonize blueschist rock surfaces. A previous report indicated that shifts in river biofilm bacterial communities could be related to the impact of WWTP outflows but these changes were suggested to be driven by the released pharmaceutical pollutants [42]. Here, we clearly show that WWTP bacterial taxa can be efficient colonizers of river compartments such as rocks and be significant contributors in the build-up of river biofilms. Even hospital-borne taxa that were highly diluted by the river flow regime could still get established among these biofilms. 

## 5. Conclusions

Wastewater treatment processes have been greatly improved over the years but still releases significant amounts of exogenous microbial cells and high concentrations of chemical pollutants. The incidence of these spillovers on natural aquatic systems remains to be investigated. Here, we have investigated the ability of WWTP clarifier sewer taxa at colonizing blueschist rocks in order to establish the likelihood of these taxa at getting established among the connected river system. This investigation was mainly performed through the use of the *tpm* metabarcoding approach which allows an allocation of DNA imprints at the species/sub-species levels for more than twenty genera. This is the first report showing that the *tpm* marker can be used to assess bacterial diversity among river and sewer systems. This approach allowed us to show that the *tpm*-profilings of blueschist biofilms generated from treated WW (of clarifiers) were highly correlated to the accumulation of certain chemical pollutants. Chemical cocktails from the hospital and domestic sewers led to distinct *tpm* bacterial communities. These differences are likely to be related to the distinct functional abilities of these taxa at using or tolerating these pollutants. 

The large spread of *tpm*-harboring bacterial taxa over the WWTP and their detection among the connected river system allowed us to apply a SourceTracker analytical scheme. This analysis showed a significant coalescence of the sewer *tpm*-harboring taxa among river epilithic biofilms. This study thus clearly demonstrated the ability of sewer taxa at colonizing inert surfaces of river systems. These taxa could harbor ARG, and be virulent and multi-resistant to antibiotics as observed for *P. aeruginosa* isolates recovered from the study site. The next step in these studies will be to investigate the evolution of resistomes and virulomes among these systems. Virulent properties might be beneficial for the colonization of highly competitive biotopes such as river rock surfaces. 

## Figures and Tables

**Figure 1 microorganisms-11-00922-f001:**
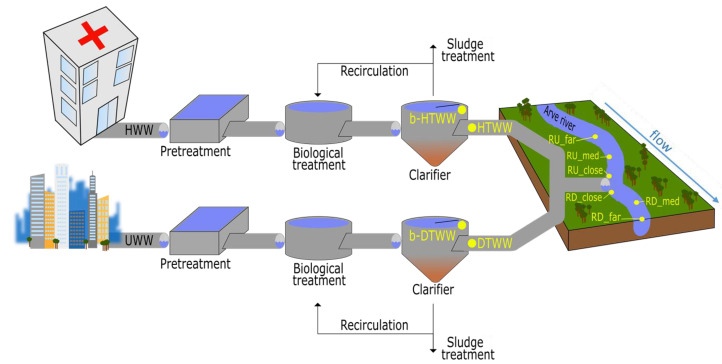
Illustration of the SIPIBEL experimental site (wastewater treatment plant of Bellecombe, Scientrier, Haute-Savoie, France). The hospital (HWW) and domestic wastewater (UWW) treatment files are shown. Treated waters (TWW) from the HTWW and DTWW clarifiers join into a single pipe prior to being delivered into the Arve river. Sampling points are represented by yellow dots. Biofilms (b) on blueschist rocks positioned 5 cm below the waterline were sampled from the hospital (b-HTWW) and domestic (b-DTWW) treatment files of the clarifiers. Treated wastewaters (HTWW and DTWW) were sampled at the outlets of the clarifiers. Blueschist biofilms (b-) and Arve River waters were sampled upstream and downstream of the outlet of the WWTP at (i) a distance of 8 km (RU_far and RD_far), (ii) 500 m (RU_med and RD_med), and (iii) 10 m (RU_close and RD_close).

**Figure 2 microorganisms-11-00922-f002:**
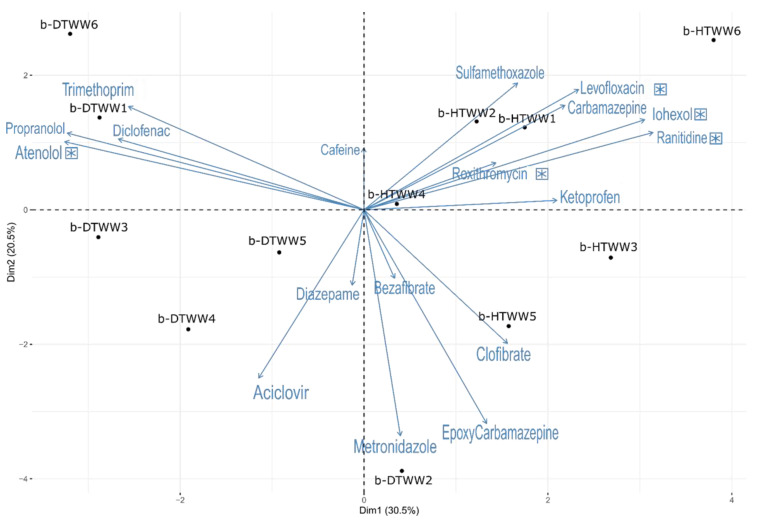
Principal component analysis (PCA) of the 18 pharmaceuticals monitored among the blueschist biofilms developing from clarifier waters (Appendix A). The PCA was computed from datasets generated from biofilms recovered from the clarifiers of the domestic (b-DTWW) and hospital (b-HTWW) treatment lines of the SIPIBEL experimental site (Figure 1). Asterisks highlight significant (Wilcoxon non-parametric tests, *p* < 0.05) differences between the monitored parameters among the b-HTWW and b-DTWW.

**Figure 3 microorganisms-11-00922-f003:**
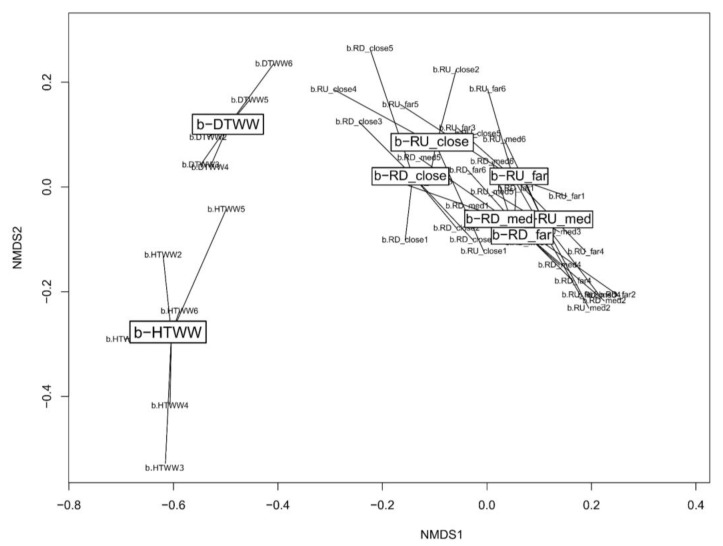
Non-metric multidimensional scaling (NMDS) representation of the Bray-Curtis dissimilarity matrix computed from the *tpm* gene ASV profiles (Appendix A) of biofilms (b-) sampled from HTWW (hospital sewer treated wastewaters), DTWW (domestic sewer treated wastewaters) and the river compartments upstream (b-RU) or downstream (b-RD) the WWTP outlet. Pairwise ANOSIM statistical tests showed significant differentiations between the *tpm* genetic profiles according to the origin of the samples (*p*-value < 0.01). River biofilm sampling sites were at 10 m from the WWTP outlet (close), at 500 m (medium (med)), and 8 km (far). See Appendix A for a full description of all samples used in this analysis.

**Figure 4 microorganisms-11-00922-f004:**
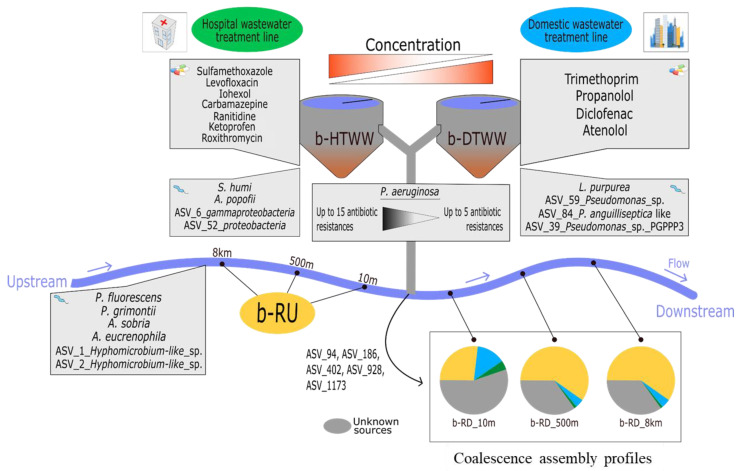
Illustration and overview of the main results observed during this study. Specific pharmaceuticals, *tpm* ASV, and inferred species from b-HTWW, b-DTWW, and river biofilms are indicated according respectively to Figure 1 and Appendix A. The coalescence of the *tpm* ASV from the b-TWW samples among the river biofilm samples are indicated according to Table 1. Color codes differentiate the compartments investigated in this study.

**Figure 5 microorganisms-11-00922-f005:**
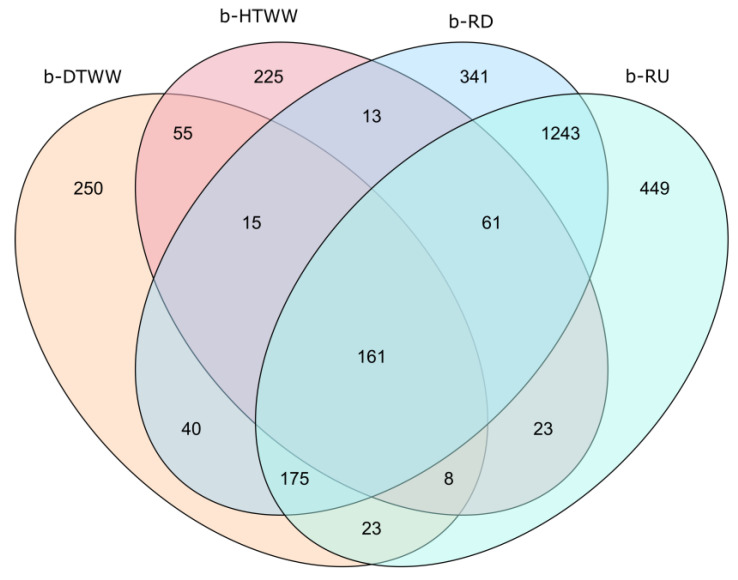
Venn diagram showing the distribution of *tpm* ASV among rock biofilms developing from domestic treated wastewaters (b-DTWW), hospital treated wastewaters (b-HTWW), and river surface water located downstream (b-RD) or upstream (b-RU) the selected WWTP outlet.

**Table 1 microorganisms-11-00922-t001:** Relative contributions of *tpm*-harboring bacterial ASV from b-HTWW (biofilms developing from hospital sewer treated wastewaters), b-DTWW (biofilms developing from domestic sewer treated wastewaters), and biofilm river taxa upstream (b-RU) the SIPIBEL WWTP outlet, in the build-up of downstream river biofilms (b-RD) as computed by a SourceTracker Bayesian analysis using datasets of Appendix A. See Appendix A for a full description of the samples.

Rock Biofilms	b-DTWW		b-HTWW		b-RU		Unknown	
	Contribution (%)	rsd (%)	Contribution (%)	rsd (%)	Contribution (%)	rsd (%)	Contribution (%)	rsd (%)
b-RD_close1	7.15	0.66	0.57	0.13	11.33	2.41	80.95	2.77
b-RD_close2	7.3	1.06	0.85	0.34	55.92	1.17	35.93	0.65
b-RD_close3	35.08	1.19	1.84	0.5	16.5	0.75	46.58	1.02
b-RD_close4	6.44	1.07	2.49	0.64	49.2	1.17	41.87	1.15
b-RD_close5	18.13	0.84	0.74	0.4	7.75	1.11	73.38	1.09
b-RD_med_1	10.87	1.08	0.44	0.22	49.24	1.19	39.45	0.71
b-RD_med_2	0.21	0.09	0.2	0.17	90.18	0.24	9.41	0.27
b-RD_med_3	1.86	0.49	0.23	0.12	81.12	0.52	16.79	0.41
b-RD_med_4	1.25	0.22	0.36	0.14	83.77	0.44	14.62	0.36
b-RD_med_5	19.28	0.75	0.89	0.52	24.54	1.26	55.29	1.5
b-RD_med_6	4.4	0.94	0.41	0.23	44.51	1.1	50.68	0.71
b-RD_far1	1.36	0.45	0.42	0.15	59.32	0.94	38.9	0.82
b-RD_far2	0.16	0.1	0.18	0.07	90	0.23	9.66	0.23
b-RD_far3	2.15	0.57	0.27	0.14	69.64	0.45	27.94	0.61
b-RD_far4	0.34	0.13	0.13	0.1	85.03	0.41	14.5	0.33
b-RD_far5	23.04	1.22	0.61	0.3	26.06	0.94	50.29	1.12
b-RD_far6	10.4	1.03	1.29	0.42	39.65	1.47	48.66	1.07
Mean	8.79	0.7	0.7	0.27	51.99	0.93	38.52	0.87

## Data Availability

The data presented in this study are available in the Appendix A of this article.

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
