# Peer review of "Evidence of Bacterial Community Coalescence between Freshwater and Discharged tpm-Harboring Bacterial Taxa from Hospital and Domestic Wastewater Treatment Plants among Epilithic Biofilms"

_microorganisms, 2023, doi:10.3390/microorganisms11040922_

Round 1

Reviewer 1 Report

Review Report of Manuscript No. microorganisms-2260854    

The manuscript entitled “Evidence of bacterial community coalescence between freshwater and discharged tpm-harboring bacterial taxa from hospital and domestic wastewater treatment plants among epilithic biofilms is quite interesting. The topic selection of the manuscript is good, in line with the current hot spots in the environmental field. In totality, this paper is pleasant to read, well-structured and well-written. However, it needs some corrections and there are some queries which the authors should kindly respond to make it good.

Some specific suggestions or questions are listed below:

1. Abstract: WWTP, please write the full name for the first time. Please check throughout the manuscript that abbreviations/acronyms are defined the first time they appear in each of three sections: the abstract; the main text; the first figure or table.

2. The Abstract should be written more precisely and explain novelty of this work.

3. Lines 59-88: Long paragraph. Please divide it into two paragraphs.

4. Introduction is easy to read but needs a little completed. I suggest this section can be shortened. Introduction should briefly place the study in a broad context and highlight why it is important. It should define the purpose of the work and its significance. Please briefly mention the main aim of the work and highlight the principal conclusions at the last of this section.

5. Line 165: COD, please define it for the first time.

6. Figure 3. Add more details in the figure legends.

7. Line 367: Candidatus Nitrospira. Please check the species names.

8. Line 393: You dont need to write italic for sp.

9. Line 468: use “Figure 5. Venn diagramm”.

10. Please check all the species names. Species names are typically given in full the first time they are used within the main text and then abbreviated throughout the remainder of the text.

11. Conclusions. This section is too long and needs improvement.  

Reviewer 2 Report

On account of the manuscript MICROORGANISMS-2260854, entitled “Evidence of bacterial community coalescence between freshwater and discharged tpm-harboring bacterial taxa from hospital and domestic wastewater treatment plants among epilithic biofilms” by Rayan Bouchali et al., the authors evaluated the ecological impact of wastewater treatment plant (WWTP)-related bacterial communities on the content of river epilithic biofilms, based on the survey for domestic treated wastewaters (DTWW) and hospital treated wastewaters (DTWW). The topic is important to better understanding of the occurrences of Pseudomonas aeruginosa and Aeromonas caviae tpm-harboring bacteria in the wastewater, and to conduct environmental risk assessment for antimicrobial resistance (AMR) in the environment as well. After careful consideration, I feel that this manuscript is to be published after improvement of some minor shortcomings. Details of my comments are as follows:

The manuscript is well written and easy to follow, and the authors got interesting results. Only minor revisions are required before publication. The present Abstract was not informative. Abstract should include purpose of the research, principal results and major conclusions in a summarized way. In addition, due to separation of the Abstract from the major article, it must be a key to lead readers to evoke a spirit of challenge to contact with the contents of the report. The authors are encouraged to improve these points for enhancement of novelty and better understanding of the results. After that I am ready to recommend the present manuscript for publication.
